# Empiric Anticoagulation Therapy in Hospitalized COVID-19 Patients: An Evaluation of Bleeding Risk Scores Performances in Predicting Bleeding Events

**DOI:** 10.3390/jcm11174965

**Published:** 2022-08-24

**Authors:** Mona A. Abdelrahman, Aya Ahmed, Abdullah S. Alanazi, Hasnaa Osama

**Affiliations:** 1Department of Clinical Pharmacy, Faculty of Pharmacy, Beni-Suef University, Beni Suef 2722165, Egypt; 2Department of Clinical Pathology, Faculty of Medicine, Beni-Suef University, Beni Suef 2722165, Egypt; 3Department of Clinical Pharmacy, College of Pharmacy, Jouf University, Sakaka 72388, Saudi Arabia; 4Health Sciences Research Unit, Jouf University, Sakaka 72388, Saudi Arabia

**Keywords:** COVID-19, anticoagulation, bleeding score, HAS-BLED score, ORBIT score

## Abstract

Currently, there is no standardized consensus on anticoagulation (AC) among patients with coronavirus disease (COVID-19), which has an overwhelming bleeding risk. We aimed to compare the patterns of AC in COVID-19 patients and compare two validated risk scores in predicting bleeding events. A retrospective review of medical records was conducted for COVID-19 patients who received empiric anticoagulation therapy. The primary outcomes included bleeding events, survival, and mechanical ventilation needs. We applied the HAS-BLED and ORBIT bleeding risk scores to assess the predictive accuracy, using c-statistics and the receiver operating curve (ROC) method. Of the included patients (*n* = 921), with a mean age of 58.1 ± 13.2, 51.6% received therapeutic AC and 48.4% received a prophylactic AC dose. Significantly higher values of d-dimer and C-reactive protein (CRP) among the therapeutic AC users (*p* < 0.001) were noted with a significantly prolonged duration of hospital stay and mechanical ventilation (*p* < 0.001 and *p* = 0.011, respectively). The mean value of the HAS-BLED and ORBIT scores were 2.53 ± 0.93 and 2.26 ± 1.29, respectively. The difference between the two tested scores for major bleeding and clinically relevant non-major bleeding was significant (*p* = 0.026 and 0.036, respectively) with modest bleeding predictive performances. The therapeutic AC was associated with an increased risk of bleeding. HAS-BLED showed greater accuracy than ORBIT in bleeding risk predictability.

## 1. Introduction

Coagulation abnormalities seem to be a challenging problem in coronavirus disease 2019 (COVID-19) [1]. Hypercoagulable states and an unusual tendency for an increased risk of thrombotic events were described among the severe and critically ill patients with COVID-19 infection [2]. The postmortem studies that assessed the mortality in COVID-19 cases showed prevalent thrombosis events as one of the main causes of death [3]. Therefore, the pathological and clinical guideline consensus reports regarding thrombosis, as a characteristic obstacle from COVID-19 in critically ill patients, recommend the need for anticoagulation in all of the COVID-19 hospitalized patients [4].

Recently, several studies have focused on extensive anticoagulation therapy to avoid the thrombosis problems [5]. Anticoagulation therapy is well-known for its associated bleeding risks; therefore, it is a treatment that needs careful dose changes and clinical monitoring [6]. The usage of anticoagulants in COVID-19 remains understudied and controversial without any definite guidelines published about the dosage, timing, and duration of anticoagulation or the optimum drug used [7].

The assessment of bleeding risk is vital in controlling patients on anticoagulation [6,8]. Two commonly used bleeding risk scores are the Hypertension, Abnormal liver/renal function, Stroke, Bleeding predisposition or history, Labile international normalized Ratio, Elderly, Drugs/alcohol (HAS-BLED) score and the Outcomes Registry for Better Informed Treatment (ORBIT) score [8,9,10]. These scores ascribe numerical valued points to the variable risk factors and collect the total points to categorize the estimated risk levels as low, intermediate, and high risk [11].

In the current study, these two bleeding risk scores, HAS-BLED and ORBIT, were evaluated for their predictive values of relevant clinical bleeding events among the hospitalized COVID-19 patients, randomized to receive prophylactic and therapeutic doses of anticoagulants.

## 2. Materials and Methods

### 2.1. Study Design and Patient Population

This observational retrospective clinical cohort study was conducted in Beni-Suef University hospital, Beni-Suef, Egypt, from 15 March to 28 December 2021. The study included adult hospitalized patients aged ≥18 years with a positive assay test of transcriptase-polymerase chain reaction (RT-PCR) for COVID-19. The patients with a negative RT-PCR, chronic users of anticoagulants before admission, patients’ records with missing data, patients with contraindications for anticoagulant therapy, or who required a suspension in anticoagulant treatment during the hospital stay were excluded from this study. This study was designed and conducted according to the Declaration of Helsinki’s ethical principles, and the study protocol was approved by the Ethics Committee of Beni-Suef University. The empirical AC indications were persistently elevated oxygen needs with a d-dimer level higher than the normal range by six times or more. Based on the AC dosage, the patients received either prophylactic anticoagulation (low-dose parenteral heparin or enoxaparin 40 mg/BID) or therapeutic anticoagulation (high-dose heparin drip, enoxaparin (1 mg/kg twice per day), apixaban (≥5 mg twice per day), or fondaparinux (≥5 mg once daily)). For the patients who met the inclusion criteria, the demographics, anticoagulation therapy dose and duration, laboratory measurements, comorbidities, and concurrent medication were collected from the medical records.

### 2.2. Bleeding Risk Scores and Outcome Measures

The bleeding risk scores, HAS-BLED and ORBIT, were assessed for all of the hospitalized patients included in the study. The HAS-BLED risk score comprised the following with one point for each: (i) uncontrolled hypertension (systolic blood pressure above 160 mmHg; (ii) stroke history; (iii) unstable/high International Normalized Ratio (INR); (iv) presence of liver/renal abnormal function; (v) prior major bleeding or predisposition; (vi) elderly > 65 years; (vii) drug predisposing to bleeding events (e.g., antiplatelet or NSAIDs); and (viii) alcohol usage. The risk scores were categorized into low at score = 0 or 1, moderate at score = 2, and high at score ≥3. The ORBIT risk score included five risk factors with one point to each of the following: age ≥75 years; compromised kidney function (eGFR < 60 mg/dL/1.73 m^2^) and antiplatelet agent use; and two points for a history of bleeding and reduced hemoglobin (less than13 mg/dL for males and less than 12 mg/dL for females). The risk scores were classified into low level (score = 0–2), moderate (score = 3), and high (score ≥ 4).

The main outcomes included major bleeding and clinically relevant non-major bleeding incidence. The secondary outcomes were disease severity progression and all-cause mortality. Major bleeding was defined in concordance with the criteria of the International Society on Thrombosis and Hemostasis (ISTH) [12], which included symptomatic and/or fatal critical bleeding in organ or area; or causing a hemoglobin level fall of about 1.24 mmol/L; or prompt blood transfusion of approximately two or more units of whole blood or red blood cells. The assessment of major bleeding was performed by two researchers. The events of clinically relevant non-major bleeding were defined as events not meeting the major bleeding criteria but associated with medical intervention or any other irritability or discomfort, such as pain or activity impairment.

### 2.3. Statistical Analysis

The categorical data were represented as frequency and percentage (%). The mean ± standard deviation (SD) described the continuous variables. The Chi-square test was applied to establish a comparison between the groups with categorical variables. The statistical analysis was performed using SPSS 16 software (Install Shield Corporation, Inc. USA, Chicago, IL, USA). The bleeding outcomes were computed by the HAS-BLED and ORBIT risk scores as the rate of bleeding events. The predictive accuracy of the bleeding risk scores for the prediction of the bleeding outcomes and all-cause mortality were estimated using the receiver operating curve (ROC) or concordance statistics (c-statistics). The statistically significant difference was set at *p*-value < 0.05.

## 3. Results

From 15 March to 28 December 2021, we screened 1362 COVID-19 patients’ medical records. Of them, 921 fulfilled the inclusion criteria and were selected for the study. The hospitalized patients included in the study received either a therapeutic (*n* = 475) or prophylactic anticoagulation dose (*n* = 446). The baseline demographic and clinical main characteristics of the study population are summarized in Table 1.

The demographics were well balanced between the groups. Overall, the study population had an average age of 58.1 years (SD 13.2), 440 (47.8%) of the patients were men, and the mean body weight was 70.47 kg (SD 12.82). A significant difference between the groups in hemoglobin, c-reactive protein (CRP), and d-dimer laboratory values was observed (*p* < 0.05). Extremely high values of CRP and d-dimer were noted among the therapeutic AC dose users (*p* < 0.001) compared to the patients on the prophylactic AC dose. Regarding the comorbidity stratification among the study population, diabetes, hypertension, chronic kidney disease, congestive heart failure, chronic liver disease, and cancer were present in 13.8%, 36.8%, 9.12%, 8.6%, 6.1%, and 2.6%, respectively, with a non-statistically significant difference between the two study groups.

The overall duration of hospital stay was mean (SD) 13.35 (3.9), with a longer duration observed for the therapeutic users with a mean (SD) value of 15.2 (3.7) days compared to the prophylactic users (mean (SD); 11.4 (3.3) days, *p* < 0.001). Additionally, the estimated rate of mortality was 7.6% and 10.7% for the patients on the prophylactic and therapeutic AC therapy, respectively. Invasive mechanical ventilation support was received by 8.6% of the study population, with a high prevalence among therapeutic AC users; however, the difference from the prophylactic AC group was statistically non-significant (*p* > 0.05). Additionally, the duration of invasive mechanical ventilation was significantly longer amongst the therapeutic dose users compared to those on prophylactic AC dosage (median; 8 and 5 days, respectively, *p*-value = 0.011).

The discrimination between the HAS-BLED and the ORBIT risk scores and their clinical usefulness assessment among the COVID-19 patients on empiric anticoagulation was performed.

Overall, 31 (3.4%) major bleeding events and 39 (4.2%) clinically relevant bleeding events were observed. The mean HAS-BLED score was 2.53 ± 0.93, with 22.15%, 57.87%, and 19.98% of patients at risk scores of low, moderate, and high, respectively. The average ORBIT score was 2.26 ± 1.29, with 47.66% of patients with low risk scores, 42.13% with moderate risk scores, and 10.21% with high risk scores. The distribution of the specific bleeding data for the HAS-BLED and ORBIT bleeding risk scores in the study cohort are depicted in Table 2.

The discriminative ability for major bleeding, as illustrated by the c-statistics analysis in ROC curve, showed a modest predictive performance; c-indexes of 0.77 in the HAS-BLED score with a 95% confidence interval [CI] of 0.69 to 0.86, *p* <0.001, and c-indexes of 0.68 in the ORBIT risk score (95% CI: 0.64 to 0.72, *p* = 0.066) (Figure 1).

For the clinically relevant non-major bleeding outcomes, the HAS-BLED c-index was 0.75 (95% CI: 0.61–0.88, *p* = 0.008) and the ORBIT score was 0.61 (95% CI: 0.44 to 0.79, *p* = 0.227) (Figure 2). A comparison of the two scores’ *c*-indexes revealed a statistically significant difference between the two tested scores for major bleeding and clinically relevant non-major bleeding (*p* = 0.026 and 0.036, respectively).

For the secondary outcome of the all-cause mortality (Figure 3), the predictive performance was poor for the two tested risk scores, with an area under the curve for the HAS-BLED risk score of 0.65 (95% CI: 0.532 to 0.785, *p* = 0.022) and a c-index of 0.45 (95% CI: 0.31 to 0.59, *p* > 0.05) for the ORBIT risk score, with non-statistical significance between the two scores (*p* = 0.62).

## 4. Discussion

Some of the studies reported increased thromboembolic incidents among the hospitalized COVID-19 patients [13,14], and enhanced outcomes with anticoagulation therapy in those populations. However, the precise impact of anticoagulation on disease management is still unclear [15,16]. Several cohort trials reported a high rate of thromboprophylaxis use among COVID-19 patients compared to those with medical illnesses other than COVID-19. In addition, the validated models for venous thromboembolism risk assessment failed to detect the patients who should have been assigned to the prophylactic anticoagulation regimens [17].

The present study investigated and evaluated the bleeding incidence rate in COVID-19 patients, using therapeutic and prophylactic anticoagulant doses for 921 critically ill patients with COVID-19, hospitalized at Beni-Suef University hospital in Egypt from 15 March to 28 December 2021.

Severe cases of COVID-19 infection were associated with hypercoagulable states, which included thrombotic events and worsened outcomes. Disseminated thrombosis and intravascular coagulopathy trigger a cytokine release and an inflammatory cascade, resulting in myocardial injury. Other proposed explanations of the cardiovascular morbidity include the direct binding of the COVID-19 virus to the angiotensin-converting enzyme-2 (ACE-2) receptors; hence, a direct invasion into the myocardial cells and hypoxia, combined with the increased metabolic demands, leads to acute myocardial injury. Cardiac injury is highly prevalent among the severe cases admitted to hospital with COVID-19 infections, with worse clinical outcomes [18,19,20]. The current evidence supports the use of prophylactic AC in the patients admitted to hospital with COVID-19 infections. Heparin was reported to exert beneficial clinical effects and reduced mortality rates compared to oral anticoagulation. Unfractionated heparin (UFH) can be used as an alternative, if low molecular weight heparin is not available. Given the state of hypercoagulopathy, higher doses of AC might be warranted for thromboprophylaxis [21,22].

The therapeutic doses of anticoagulation therapy were associated, in our study, with a significant elevation in the d-dimer levels, CRP, and bleeding events. This attitude reflects the repetition of anticoagulation in COVID-19 patients because of the infection severity. The elevated d-dimer level has been related to COVID-19 disease severity and increased death risk [23]. d-dimer may help in the prediction of intravascular coagulation and micro-thrombosis, which is a major determinant of the ambiguous respiratory damage [24]. The raised d-dimer and CRP levels may reveal wide proteolysis and fibrinolysis activity of the plasmin through matrix metalloproteinase activation, which causes tissue injury and inflammation [25]. In addition to the typical finding of d-dimer elevation among the therapeutic AC users, in this trial and consistent with other studies [26,27], we observed a modest non-significant decrease in the estimated platelet count. Taken together, the thrombocytopenia and elevated d-dimer suggest disseminated coagulopathy; however, the pattern might differ from sepsis-related coagulopathy. In fact, sepsis-related thrombocytopenia is usually highly profound, and the d-dimer levels are beyond those observed in COVID-19 patients. Thus, most of the hospitalized COVID-19 patients with disseminated intravascular coagulopathy (DIC) would not fulfill the validated DIC score [22], which might distract the coagulation scheme and can augment the bleeding risk among COVID patients [28,29].

In the present study, the mortality rate for the patients on therapeutic AC regimens was 10.7%, with a longer duration of hospital stay, compared to 7.6% for the patients on prophylactic therapy. Therefore, the therapeutic AC in the present study was linked to increased in-patient mortality. Unlike our results, anticoagulation therapy has been linked to increased survival in severe COVID-19 infections. In agreement with our results, Musoke, N. et al. described that the therapeutic dose of AC increased the burden of mortality in the hospitalized COVID-19 patients [6]. However, a recent study showed the beneficial effects of anticoagulation therapy among the patients with severe infections and a high risk of thrombosis [30]. The discrepancy from our results can be explained by the fact that other factors may have affected the mortality rate of patients in our study, such as disease severity, comorbidities, and major complications. This might also be a result of selection bias, since the severe cases with deteriorating d-dimers and elevated levels of inflammatory markers were assigned to therapeutic anticoagulation.

In addition, COVID-19 disease is characterized by diffuse alveolar damage resulting from micro-vascular blockage in the lung circulation, which might hinder gas exchange, leading to the significant hypoxemia detected in these patients, who then need non-invasive mechanical ventilation [31]. In the same way, this study found that the duration of invasive mechanical ventilation of therapeutic dose users was significantly longer compared to those on prophylactic AC dosage amongst the hospitalized COVID-19 patients. The same was reported in an observational study conducted at Latifa hospital, Dubai by Elmelhat et al. [2]. This study found that there was a longer duration of hospital stay, mechanical ventilation and also elevated dimer levels with the therapeutic doses of AC therapy. On the contrary, Lemos et al. described opposite findings, where they reported that the usage of a therapeutic dose of enoxaparin led to a gas exchange improvement, enhanced d-dimer levels, and afterwards, advanced levels of effective weaning from ventilators in the critically ill COVID-19 patients with respiratory distress [31]. However, the Lemos et al. trial only involved mechanically ventilated patients with age limits that did not exceed 85 years, a d-dimer level greater than 1000 μg/L, and without severe damage to the organs before the infection (class III or IV Heart failure, Child–Pugh B or C cirrhosis, etc.) or patients who could not tolerate AC [30]. This may be also due to the fact that the increase in the ratio of the oxygen saturation to the inspired oxygen (SpO_2_/FiO_2_) fraction was a vital predictive indicator for COVID-19 patients, as demonstrated by Lu et al. [29]. That is also in line with the study by Lemos A.C. et al., which reported a significant progress in the PaO_2_/FiO_2_ ratio with the use of therapeutic enoxaparin compared to the prophylactic doses [31].

The net outcomes of anticoagulation therapy may vary depending on the onset of initiation throughout the disease course, as well as coagulation and inflammation severity. In this trial, and consistent with the REMAP-CAP Investigators trial, although the numerical value of the major bleeding events was higher with therapeutic AC compared to prophylactic AC, it was still a relatively low rate (4%). The autopsies of COVID-19 patients with severe acute respiratory distress syndrome revealed micro-thrombus formation, in addition to hemorrhages in the alveoli. As a result, with severe pulmonary inflammation, therapeutic AC might worsen the alveolar hemorrhages and the clinical outcomes [32].

We observed a significant drop in the hemoglobin levels in the therapeutic group compared to the prophylactic group with clinical evidence of bleeding in those patients. The inflammatory process may explain this anemia as the cytokines decrease the erythropoietin levels, disturb the iron homeostasis by several mechanisms, directly prevent the proliferation and differentiation of erythroid cells, and diminish the survival of the red blood cells [33]. The bleeding events were higher among the therapeutic users of AC for major and clinically relevant non-major bleeding events; however, the difference was non-statistically significant for the major bleeding events’ rate. Although the major bleeding events’ rate was small, this should be a warning sign for the potential risk of AC hemorrhage complications, which are linked with high mortality rates and even long-term morbidity among the survivors.

The potential benefits of AC in preventing thrombosis need to be balanced with its risks of adverse events, especially bleeding. Accordingly, the HAS-BLED and ORBIT scores may offer hopeful tools to measure the bleeding risk.

In this study, the HAS-BLED and ORBIT scores were used in a clinical cohort trial of hospitalized COVID-19 patients; we showed that the HAS-BLED scoring system more precisely predicted clinically relevant bleeding events when compared with the ORBIT score. We opted to assess these two scores since both of the scores, particularly the HAS-BLED score, are considered to be the most commonly applied scoring systems in trials and clinical practice [8,9].

The application of these bleeding risk scores can be beneficial in clinical practice by drawing attention to bleeding risk factors that could be modified; hence; may alleviate or eliminate the potential risk. Our analysis consistently confirmed that a higher risk score in the HAS-BLED system was associated with the evaluated outcomes: major bleeding; non-major bleeding; and all-cause mortality. The association between bleeding events and in-hospital mortality is well documented, consistent with our results, where the HAS-BLED score was related to bleeding events and, thus, to mortality [34,35].

We recommend that the application of the HAS-BLED assessment system can be helpful in risk stratification and controller AC dose adjustment. Therefore, the HAS-BLED score can be utilized for clinical or electronic alerts to ‘flag up’ patients with a potential risk of bleeding for more careful evaluation and medical follow-up and to draw more attention to potential bleeding risk factors that can be modified [36].

The recent evidence concerning the therapeutic anticoagulation dose benefits in COVID-19 management is varied, with particular studies presenting possible benefits while others, such as the current study, demonstrating bleeding risk and mortality. As promising as it is, an objective assessment of the bleeding scores before the beginning anticoagulation therapy would provide a way to accurately weigh up the potential benefits and risks. Wider forthcoming trials are needed to actually control the degree of benefits and risks of anticoagulation in COVID-19 patients.

However, this study has some limitations. First, although an increased risk of bleeding was observed in this study with therapeutic AC, we did not estimate the predictive benefit of AC on the ischemic risks. Second, the retrospective single-center design of the study may limit the generalizability of the study to a different population. Third, some of the outcomes may not be recorded properly because of the retrospective design of the study. Fourth, in this study, the risk scores were calculated once without estimating the possibility of the dynamic changes of the risk scores during the follow-up, which may affect our findings.

## 5. Conclusions

Therapeutic AC is associated with a high bleeding risk y which, as a consequence, is associated with increased mortality rates. The balance between the benefits and risks of anticoagulation should be taken into consideration for COVID-19 patients. For clinical utility in prediction of bleeding events, the HAS-BLED was more beneficial compared to the ORBIT scoring system. The use of a feasible and accurate risk score, such as the HAS-BLED, can play a pivotal role for bleeding risk evaluation and ultimately minimize the incidence of bleeding events.

## Figures and Tables

**Figure 1 jcm-11-04965-f001:**
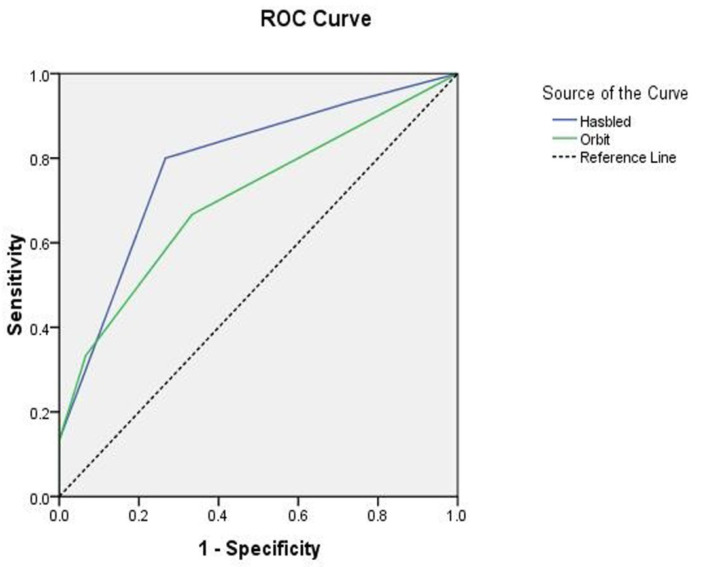
The receiver operating curve (ROC) of the two risk scoring systems, HAS-BLED and ORBIT, in predicting major bleeding events among the COVID-19 patients.

**Figure 2 jcm-11-04965-f002:**
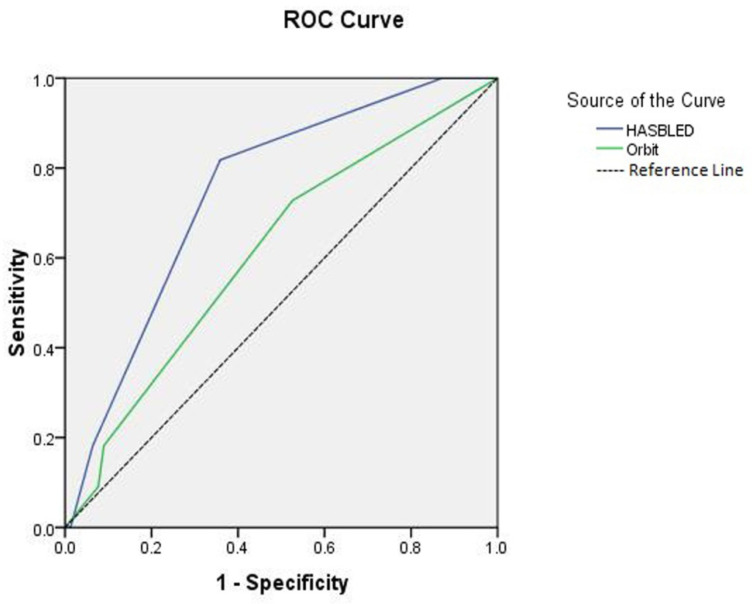
The receiver operating curve (ROC) of the two scores, HAS-BLED and ORBIT, in predicting clinically relevant non-major bleeding events among the COVID-19 patients.

**Figure 3 jcm-11-04965-f003:**
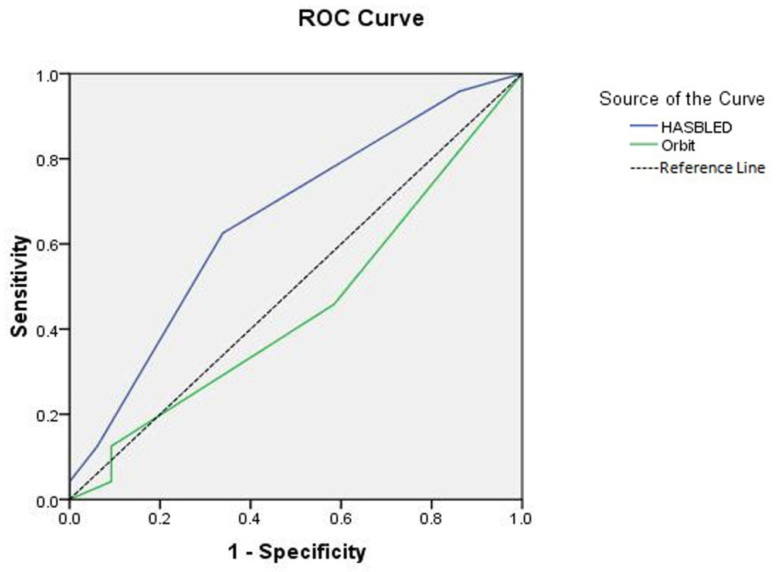
The receiver operating curve (ROC) of the two scores, HAS-BLED and ORBIT, in predicting the all-cause-mortality among the COVID-19 patients.

**Table 1 jcm-11-04965-t001:** Baseline demographic and clinical data.

Characteristics	Total	Prophylactic AC	Therapeutic AC	*p*-Value
	*n* = 921	*n* = 446	*n* = 475	
**Age (mean ± SD)**	58.1 ± 13.2	57.8 ± 13.1	58.5 ± 13.4	0.441
**≤60 years, *n* (%)**	489 (53.1)	247 (55.4)	242 (50.9)	
**>60 years, *n* (%)**	432 (46.9)	199 (44.6)	233 (49.1)	
**Gender, *n* (%)**				
**Male**	440 (47.8)	201 (45.1)	239 (50.3)	0.112
**Female**	481 (52.2)	245 (54.9)	236 (49.7)	
**Body weight**	70.47 ± 12.82	69.64 ± 11.67	71.25 ±13.79	0.057
**Comorbidities, *n* (%)**				
**Diabetes**	127 (13.8)	54 (12.1)	73 (15.4)	0.091
**Hypertension**	339 (36.8)	161 (36.1)	187 (39.4)	0.173
**Chronic kidney disease**	81 (9.12)	32 (7.2)	49 (10.3)	0.052
**Chronic liver disease**	79 (8.6)	35 (7.8)	44 (9.3)	0.223
**Congestive heart failure**	56 (6.1)	22 (4.9)	34 (7.16)	0.082
**Cancer**	24 (2.6)	13 (2.9)	11 (2.3)	0.283
**Concurrent medications, *n* (%)**				
**Antiplatelet**	148 (16.1)	71 (15.9)	77 (16.2)	0.451
**NSAIDs**	89 (9.7)	38 (8.5)	51 (10.7)	0.129
**Systemic steroid therapy**	36 (3.9)	14 (3.14)	22 (4.63)	0.119
**Laboratory measurements (mean ± SD)** **Hemoglobin, g/dL** **C-reactive protein, mg/dL** **d-dimer, ng/mL** **Platelet count, 10^9^/L**				
	11.78 ± 5.49	12.29 ± 5.73	11.32 ± 5.21	0.007 *
	91.4 ± 39.6	66.1 ± 24.8	115.2 ± 36.1	<0.001 *
	773.1 ± 522.7	360 ± 142.6	1161 ± 452.1	<0.001 *
	251.03 ± 77.5	254.7 ± 84.3	247.6 ± 70.5	0.165
**Risk scores**				
**HAS-BLED risk score (mean ± SD)**	2.53 ± 0.93	2.49 ± 0.97	2.56 ± 0.89	0.253
**ORBIT risk score (mean ± SD)**	2.26 ± 1.29	2.18 ± 1.3	2.33 ± 1.27	0.077
**Clinical outcomes**				
**Invasive mechanical ventilation, *n* (%)**	79 (8.6)	32 (7.17)	47 (9.89)	0.158
**Duration of mechanical ventilation, days median (IQR)**	5.5 (5–8)	5 (4–6)	8 (5–10)	0.011 *
**Major bleeding events, *n* (%)**	31 (3.4)	12 (2.7)	19 (4)	0.137
**Clinically relevant non-major bleeding, *n* (%)**	39 (4.2)	13 (2.9)	26 (5.5)	0.025 *
**Duration of hospital stay, days**	13.35 ± 3.9	11.4 ± 3.3	15.2 ± 3.7	<0.001 *
**All-cause mortality, *n* (%)**	85 (9.2)	34 (7.6)	51 (10.7)	0.052

(*) denotes statistically significant at *p* < 0.05.

**Table 2 jcm-11-04965-t002:** Bleeding events and mortality incidence among trial population stratified according to the HAS-BLED and ORBIT risk scores.

Risk Scores	Total	Clinically Relevant Non-Major Bleeding	Major Bleeding	All-Cause Mortality
	*n* (%)	*n* (%)	*n* (%)	
**HAS-BLED**				
**Low (0–1)**	204 (22.15)	10 (1.1)	6 (0.65)	17 (1.8)
**Moderate (2)**	533 (57.87)	12 (1.3)	10 (1.1)	28 (3.04)
**High (≥3)**	184 (19.98)	17 (1.8)	15 (1.6)	40 (4.3)
**ORBIT**				
**Low (0–2)**	439 (47.66)	14 (1.5)	8 (0.86)	29 (3.1)
**Moderate (3)**	388 (42.13)	13 (1.4)	12 (1.3)	25 (2.7)
**High (≥4)**	94 (10.21)	12 (1.3)	11 (1.2)	31 (3.4)

## Data Availability

The datasets used and/or analyzed during the current study are available from the corresponding author on reasonable request.

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
