# Peer review of "Empiric Anticoagulation Therapy in Hospitalized COVID-19 Patients: An Evaluation of Bleeding Risk Scores Performances in Predicting Bleeding Events"

_jcm, 2022, doi:10.3390/jcm11174965_

Round 1
Reviewer 1 Report
In the article entitled “Empiric anticoagulation therapy in hospitalized COVID-19 patients: an evaluation of bleeding risk scores performances in predicting bleeding events” M.A. Abdelrahman et al retrospectively analyzed medical records of COVID-19 patients receiving anticoagulation in order to compare the bleeding incidence rate using therapeutic or prophylactic anticoagulant doses and to evaluate HAS-BLED and ORBIT risk scores in predicting bleeding events. They found that the difference between the 2 tested scores for major bleeding and clinically relevant non-major bleeding was significant (p=0.026, and 0.036, respectively) with a modest bleeding predictive performances. HAS-BLED was beneficial compared to ORBIT score in prediction of bleeding events. They also found that significantly higher values of D-dimer and C-reactive protein (CRP) among therapeutic AC users (p<0.001) were noted with significant prolonged duration of hospital stay and mechanical ventilation (p<0.001, and p=0.011, respectively). The topic is surely interesting and overall well-written. However, some issues have to be addressed.
Major comments:
1) The major issue of this analysis is that the authors evaluated bleeding risks, concluding that therapeutic AC was associated with an increased risk of bleeding (as it could be surely anticipated) but they do not weigh this disadvantage with a predictive benefit on the ischemic risks. Indeed, is now well-known that COVID-19 may lead to endothelial dysfunction, systemic inflammatory response ultimately resulting in micro- and macro-vessels thrombosis. I understand that the authors an extensive discussion on the ischemic risks related to COVID-19 are beyond the scope of this work. Nevertheless, this significant limitation should be included in the limitation section. In the discussion section, myocardial damage due to ischemic issues should be discussed as well, mentioning coronary artery involvement (as highlighted in these early reports on JCM, as highlighted in these studies “Redefining the Prognostic Value of High-Sensitivity Troponin in COVID-19 Patients: The Importance of Concomitant Coronary Artery Disease - J. Clin. Med. 2020, 9, 3263; doi:10.3390/jcm9103263” – “Prognostic significance of cardiac injury in COVID-19 patients with and without coronary artery disease. Coron Artery Dis. 2020. doi:10.1097/MCA.0000000000000914.” – “The relationship between coronary artery disease and clinical outcomes in COVID-19: a single-center retrospective analysis. Coron Artery Dis. 2020 Jul 23. doi:10.1097/MCA.0000000000000934.”
2) Also, in relation to point #1, potential advantages/disadvantages of AC therapy, should be further discussed. Indeed, not all AC are born equal, and VKA/OAC/heparin may show different anticoagulant and anti-ischemic properties, besides of their direct mechanism of action. This should be mentioned in the discussion section; e.g. mention the pleiotropic effect that for example heparin may show, in relation to these studies: Coagulation abnormalities and thrombosis in patients with COVID-19. Lancet Haematol. 2020 doi: 10.1016/s2352-3026(20)30145-9. --- Prevalence of venous thromboembolism in patients with severe novel coronavirus pneumonia. J. Thromb. Haemost. 2020;18:1421–1424. doi: 10.1111/jth.14830. --- Oral anticoagulation and clinical outcomes in COVID-19: An Italian multicenter experience. Int J Cardiol. 2020 Sep 8:S0167-5273(20)33735-9. doi: 10.1016/j.ijcard.2020.09.001 --- pulmonary vascular endothelialitis, thrombosis, and angiogenesis in Covid-19. N. Engl. J. Med. 2020 doi: 10.1056/nejmoa2015432).
3) In the conclusion section the authors state: “Randomized-controlled trials are required to evaluate the clinical implications of anticoagulation in COVID-19 patients.” This conclusion does not appear entirely correct since, to date, some RCT on this topic have already been published: see in particular REMAP-CAP Investigators. Therapeutic Anticoagulation with Heparin in Critically Ill Patients with Covid-19. N Engl J Med. 2021 Aug 26;385(9):777-789. doi: 10.1056/NEJMoa2103417. All RCT on this topic should be discussed.
4) “This elevation of the D-dimer in COVID-19 patients may not essentially directly reproduce thrombotic threat but relatively a disease severity”. Please clarify this sentence, since these findings should be analyzed very carefully. D-dimer elevation only rarely indicates DIC in COVID-19, which is however followed by specific clinical signs and symptoms. Indeed, D-dimer elevation may sometimes predict a vascular involvement, coagulopathy and or silent microthrombosis, but it is still difficult to understand if it is an INDEPENDENT predictor of disease severity or just a bystander of an underlying condition that may worsen COVID-19 prognosis. For the same reason, D-dimer elevation does not imply a mandatory anticoagulant treatment, which would be otherwise compulsory in overt DIC cases, although heparin administration was found to be beneficial only in patients with high D-dimer levels and in case of sepsis-induced coagulopathy in this work (Tang, H et al., Anticoagulant treatment is associated with decreased mortality in severe coronavirus disease 2019 patients with coagulopathy, J. Thromb. Haemost. (2020) - doi.org/10.1111/jth.14817). Moreover, findings about D-dimer elevation could be better discussed; in other articles it has been reported that, although not being an independent marker of disease severity, it may sometimes predict a vascular involvement, coagulopathy and or silent microthrombosis. Microthrombosis appeared to be the major determinant of an unexplained respiratory damage, as described here: “Prevalence and outcome of silent hypoxemia in COVID-19. Minerva Anestesiol. 2021 Mar;87(3):325-333. doi: 10.23736/S0375-9393.21.15245-9. PMID: 33694360.” Please mention this topic in the discussion section. Finally, if it is true that D-dimer elevation is somehow capable to predict worse outcomes, its role should be further analyzed as well as subsequent anticoagulant therapy in most severe COVID-19 cases. Please discuss in details your findings on this topic.
Minor comments:
1) There some minor language/syntaxis errors that should be fixed with a language revision of the entire manuscript.
Reviewer 2 Report
Interesting manuscript, well supported with clear discussion. I appreciated the synthetic style of work development that helps to remind presented data. The paper is well written and formally correct and it is clear its clinical relevance, and what this article should add to the body of knowledge on this topic. Mumoli N, Conte G, Cei M, Vitale J, Capra R, Rotiroti G, Porta C, Monolo D, Colombo A, Mazzone A, Kucher N, Konstantinides SV, Dentali F, Barco S. In-hospital fatality and venous thromboembolism during the first and second COVID-19 waves at a center opting for standard-dose thromboprophylaxis. Thromb Res. 2021 Jul;203:82-84.
